# HSC70-3 in the Gut Regurgitant of Diamondback Moth, *Plutella xylostella*: A Candidate Effector for Host Plant Adaptation

**DOI:** 10.3390/insects16050489

**Published:** 2025-05-02

**Authors:** Qingxuan Qiao, Chanqin Zheng, Huiting Feng, Shihua Huang, Bing Wang, Uroosa Zaheer, Weiyi He

**Affiliations:** 1State Key Laboratory of Agricultural and Forestry Biosecurity, Institute of Applied Ecology, Fujian Agriculture and Forestry University, Fuzhou 350002, China; qqx177845@gmail.com (Q.Q.); w1079754421@163.com (C.Z.); feng06092022@126.com (H.F.); huangshihua@fafu.edu.cn (S.H.); wangbing057@163.com (B.W.); uroosazaheer58@gmail.com (U.Z.); 2International Joint Research Laboratory of Ecological Pest Control, Ministry of Education and Ministerial and Provincial Joint Innovation Centre for Safety Production of Cross-Strait Crops, Fujian Agriculture and Forestry University, Fuzhou 350002, China

**Keywords:** *Plutella xylostella*, HSC70-3, effector, gut regurgitant, host adaptation

## Abstract

Plants and herbivorous insects have co-evolved in a continuous arms race, with plants developing complex defense systems and insects evolving strategies to circumvent them. The diamondback moth (*Plutella xylostella*), a major global pest of cruciferous crops, has developed effective mechanisms to adapt to its host plants. In this study, we identified a shock cognate protein, HSC70-3, in the gut regurgitant of *P. xylostella*, which accumulates at the feeding sites on host plants. Our results suggest that this protein may suppress plant defense responses, thereby facilitating insect survival and development. Notably, when the gene encoding HSC70-3 was knocked out using CRISPR/Cas9, larvae exhibited significantly reduced growth and pupal weight on the radish (*Raphanus sativus*) host plants, while no such effects were observed on an artificial diet containing rich protein and carbohydrates. This indicates that HSC70-3 is specifically involved in mediating insect adaptation to plant-derived challenges rather than basic nutritional processes. The findings provide new insight into the role of insect proteins in shaping plant–insect interactions and suggest potential directions for improving crop protection by targeting specific insect effectors.

## 1. Introduction

Herbivorous insects, with nearly 500,000 identified species, serve as a major driver of biodiversity and a fundamental component of terrestrial ecosystems [1,2]. Over 400 million years of co-evolution, plants and herbivorous insects have been engaged in a continuous arms race, driving the evolution of intricate plant defense mechanisms and corresponding insect adaptations that promote survival [3,4,5]. To counteract herbivory, plants have evolved sophisticated chemical and structural defenses, including toxic secondary metabolites, proteinase inhibitors, and immune signaling pathways [6,7,8]. In turn, herbivorous insects have evolved strategies to circumvent or suppress these defenses, enabling efficient host plant utilization [9,10,11,12]. During feeding, the insect herbivores deposit oral secretions (OS) onto wounded plant tissues, where they interact with plant signaling pathways and modulate defense responses [13,14]. The regurgitant originating from the gut contents is the primary component of OS and carries digestive enzymes, microbial communities, and metabolic compounds that further modulate plant responses [15]. These secretions facilitate digestion, detoxification, and immune modulation, ultimately promoting insect adaptation to host plants [16,17].

Previous findings highlight the essential roles of effectors in overcoming plant defenses and facilitating insect adaptation [18,19,20]. For example, Glucose oxidase (GOX), a well-characterized effector secreted by the labial glands of caterpillars such as *Helicoverpa zea*, suppresses nicotine biosynthesis and other defense-related compounds in tobacco while attenuating the jasmonic acid (JA) signaling pathway through the upregulation of salicylic acid (SA) signaling [18]. Notably, *H. zea* larvae feeding on tomato (*Solanum lycopersicum*) induce the expression of the proteinase inhibitor gene *Pin2* and elevate JA levels, whereas feeding on tobacco (*Nicotiana tabacum*) increases SA levels. This differential response indicates that GOX, as a salivary-gland-derived effector, may exert distinct functions depending on the host plant species [19]. Interestingly, GOX has also been identified in the regurgitant of *P. xylostella*, where it catalyzes the oxidation of plant-derived glucose, leading to the production of H_2_O_2_. The accumulation of H_2_O_2_ activates the SA pathway while suppressing the JA pathway, thereby attenuating JA-mediated defense responses against chewing insects [20]. However, the transcription and enzymatic activity of GOX can be suppressed by specific bacterial strains isolated from the regurgitant of *P. xylostella*, thereby impairing its suppression of plant defense responses and negatively influencing the insect’s adaptation to host-induced immunity [21]. Another representative effector, *Helicoverpa armigera* R-like protein 1 (HARP1), identified in the OS of *H. armigera*, is structurally homologous to an R-like protein found in the venom glands of *Nasonia vitripennis* (parasitoid wasp). HARP1 infiltrates host cells through feeding-induced plant wounds, where it interacts with JASMONATE-ZIM-domain (JAZ) proteins to suppress JA signaling and attenuate plant defenses. Notably, *harp1* transcripts are highly expressed in the foregut and midgut rather than in the labial glands, and the protein is detectable in both tissues, underscoring its function in host plant adaptation through regurgitation [22]. These findings have also been shown in other insect species [23], further highlighting the role of regurgitant-associated effectors in enhancing the adaptation of chewing insects to host plants.

Although well-established effectors such as GOX and HARP1 have been extensively studied for their role in modulating plant defenses and facilitating insect adaptation to host plants, heat shock proteins (HSPs) have received comparatively less attention as potential effectors in insect–plant interactions. As molecular chaperones, HSPs facilitate protein folding, stabilization, and transmembrane transport, thus contributing to cellular adaptation to both biotic and abiotic stressors [24,25]. HSP70 is particularly noteworthy due to its high evolutionary conservation, widespread presence across various organisms, and diverse functional roles [26,27,28]. Traditionally known for their role in maintaining cellular homeostasis and mediating stress responses [29,30,31], HSP70s are increasingly acknowledged for their potential role in modulating plant immunity during insect feeding. Notably, studies have shown that insect HSP70s, including *Nilaparvata lugens* NlHSC70-3, have been shown to suppress plant immune responses by downregulating defense-related genes and reducing reactive oxygen species (ROS) accumulation [32]. In contrast, *NlDNAJB9*, an HSP40 family member in *N. lugens*, triggers ROS bursts and activates immune responses through its DNAJ domain, thereby enhancing plant resistance to both insect herbivory and pathogens [33]. These findings highlight the dual potential of HSPs in plant–insect interactions, where they can both suppress plant immune responses and, under certain conditions, activate plant defense mechanisms. Given the potential roles of HSP proteins in insect–plant interactions, their functions in most herbivorous insects remain largely unexplored, particularly in the cosmopolitan pest *Plutella xylostella*, which is of significant economic importance due to its widespread infestation of cruciferous crops worldwide and its ability to rapidly adapt to various host plants.

In this study, we identified HSC70-3 in the gut regurgitant of *P. xylostella*. Additionally, it is secreted into plant wound sites during larval feeding. Short-term host transfer experiments revealed dynamic tissue-specific regulation of *hsc70-3*, indicating its differential regulation at transcriptional and post-translational levels in response to host plant shifts and suggesting its role in plant-induced stress responses and host adaptation. Furthermore, the knockout of *hsc70-3* resulted in impaired larval growth, prolonged development, and reduced pupal weight on host plants, highlighting its role in host adaptation. These findings provide initial evidence that HSC70-3 functions as an effector in *P. xylostella* interactions with host plants, playing a crucial role in suppressing plant defenses.

## 2. Materials and Methods

### 2.1. Insect Strains and Rearing Conditions

The *P. xylostella* strain used in this study was obtained from the Institute of Zoology, Chinese Academy of Sciences (Beijing, China), and has been maintained under insecticide-free and selection-free conditions since 2016 [34]. The larvae were reared on an artificial diet (AD) under controlled conditions (25 ± 2 °C, 65 ± 5% relative humidity, and a 16:8 h light-dark photoperiod). Larvae were reared in 150 mm × 150 mm × 80 mm cardboard containers, and the diet was replenished every three days to ensure optimal development. The artificial diet was prepared by dissolving agar (12 g) and yeast powder (40 g) in 500 mL of distilled water with continuous stirring. After cooling to approximately 50 °C, the remaining ingredients were added, including wheat germ powder (75 g), vitamin powder (2 g), sorbic acid (2 g), ethyl *p-hydroxybenzoate* (2 g), ascorbic acid (2 g), radish seed powder (6 g), sucrose (20 g), linoleic acid (0.2 mL), and rapeseed oil (2 mL), followed by thorough mixing [34]. The homogenized mixture was poured into sterilized trays and allowed to solidify before use. Once pupation occurred, pupae were collected and transferred to 100 mm × 100 mm × 100 mm containers for adult emergence and mating. To promote oviposition, adult moths were provided with a 10% honey solution as a nutritional supplement.

### 2.2. Plant Materials and Growth Conditions

The radish variety ‘Nanpan Prefecture’ used in this experiment was provided by Kaiyisun Biotechnology Co., Ltd. (Shanghai, China) and was stored under dry conditions before sowing. Plants were grown in 450 mm × 300 mm × 70 mm plastic trays filled with a 1:1 (*v*/*v*) mixture of peat soil and vermiculite without additional fertilization to maintain controlled growth conditions. Trays were watered with sterile distilled water every two days to maintain optimal soil moisture. All plants were cultivated in a climate-controlled growth chamber (ZYL202310001, Fuzhou, Fujian Zhongyouling Technology Co., Ltd., China), under a 16:8 h light/dark cycle, 2000 lux light intensity, 23 ± 1 °C, and 65 ± 5% relative humidity. Radish seedlings at the fully expanded cotyledon stage, defined by the presence of two fully expanded cotyledons without visible true leaves (~7 days post-germination), were selected for insect feeding experiments.

### 2.3. Short-Term Host Transfer Experiment

The short-term host transfer experiment involved transferring first-instar *P. xylostella* larvae from an artificial diet to radish seedlings. The rearing conditions for the larvae on radish seedlings were identical to those on the artificial diet. Larvae were reared on the host plant for two weeks, during which fourth-instar individuals were collected for tissue dissection. The same procedure was followed for *hsc70-3* mutant larvae, with the larvae transferred to the host plant for the same duration under identical rearing conditions.

### 2.4. Sample Collection, RNA Extraction, cDNA Synthesis, and Gene Cloning

Samples from various developmental stages of *P. xylostella* were collected, including eggs (E) (*n* = 600), first-instar larvae (L1) (*n* = 7 for each replicate), second-instar larvae (L2) (*n* = 7), third-instar larvae (L3) (*n* = 7), fourth-instar larvae (L4) (*n* = 7), pupae (P) (*n* = 7), and adults (A) (*n* = 7) (sex not differentiated). Fourth-instar larvae were rinsed in phosphate-buffered saline (PBS) and dissected under sterile conditions to minimize contamination. Each larva was placed in a Petri dish containing 500 µL sterile PBS and dissected under a stereomicroscope (SZ660, Chongqing Aote Optical Instrument Co., Ltd., Chongqing, China) using sterilized fine forceps (Dumont, Montignez, Switzerland). The dissected tissues, including the fat body (FB), gut contents (GCT), head (HD), Malpighian tubules (MT), gut (GUT), silk gland (SK), and remaining tissues (RT), were carefully isolated and washed twice with PBS to remove residual debris. For each experimental group, twenty remaining tissue samples were collected, while 50 larvae were used for each of the other tissue types. The collected tissues were transferred into separate RNase-free microcentrifuge tubes and immediately flash frozen in liquid nitrogen. Samples were stored in an ultra-low temperature freezer (MDF-U54V, Panasonic, Osaka, Japan) until further use, with three biological replicates per tissue type, each consisting of pooled tissues from multiple individuals.

Total RNA was isolated from *P. xylostella* using the Total RNA Extraction Kit (Promega, Madison, WI, USA) following the manufacturer’s protocol. RNA purity and concentration were measured using a NanoDrop™ 2000 spectrophotometer (Thermo Scientific, Waltham, MA, USA) with A260/A280 ratios between 1.8 and 2.2 deemed acceptable. RNA integrity was assessed via 1% agarose gel electrophoresis, with only high-quality RNA samples selected for subsequent experiments.

First-strand cDNA was synthesized using the FastKing RT Kit (including gDNase treatment) (TIANGEN Biotech, Beijing, China). Each 20 μL reaction mixture consisted of 4.0 μL HiScript III enzyme mix, 14 μL RNase-free ddH_2_O, and 2 μL total RNA (500 ng–1 μg). To ensure efficient cDNA synthesis, the reaction mix included random hexamer primers and oligo(dT)_18_ primers. The reaction was carried out at 42 °C for 15 min, followed by heat inactivation at 95 °C for 3 min using a thermal cycler (S1000 Thermal Cycler, Bio-Rad Laboratories, Hercules, CA, USA).

The *hsc70-3* gene sequence of *P. xylostella* was identified from gut regurgitant transcriptomic data and retrieved from the DBM Genome Database (http://iae.fafu.edu.cn/DBM/, accessed on 1 March 2025) and validated via NCBI BLAST online tool (https://blast.ncbi.nlm.nih.gov/Blast.cgi, accessed on 1 March 2025) [35]. Gene-specific primers for *hsc70-3* amplification were designed using Primer Premier software (version 6.0; PREMIER Biosoft, Palo Alto, CA, USA) (Table A1). PCR was performed with 2× Rapid Taq Master Mix (Vazyme Biotech, Nanjing, China) in a 20 μL reaction mixture, consisting of 10 μL of 2× Master Mix, 0.5 μL of forward primer (10 μM), 0.5 μL of reverse primer (10 μM), 1 μL of cDNA template, and 8 μL of RNase-free ddH_2_O. The PCR cycling conditions were as follows: initial denaturation at 95 °C for 3 min, followed by 35 cycles of 95 °C for 15 s, 60 °C for 15 s, and 72 °C for 30 s, with a final extension at 72 °C for 10 min.

The *hsc70-3* PCR product was purified using the Gel Extraction Kit (Omega Bio-Tek, Norcross, GA, USA) and subsequently ligated into the pJET1.2/blunt vector using the CloneJET PCR Cloning Kit (Thermo Fisher Scientific, Waltham, MA, USA). The ligation reaction was conducted at 22 °C for 2 h in a total volume of 10 μL, consisting of 5 μL 2× Reaction Buffer, 2 μL purified PCR product, 1 μL T4 DNA ligase, 1 μL pJET1.2/blunt Cloning Vector, and 2 μL RNase-free ddH_2_O. The ligation mixture was transformed into chemically competent *Escherichia coli* DH5α cells via the heat-shock method (42 °C for 45 s, followed by immediate cooling on ice for 2 min). Transformed cells were spread onto LB agar supplemented with 100 μg/mL ampicillin and incubated at 37 °C overnight. Following incubation, colony PCR was performed using vector-specific primers to screen for positive clones. Six independent positive clones were selected and sent to Sangon Biotech (Shanghai) Co., Ltd. for Sanger sequencing to verify the presence and accuracy of the inserted sequences.

### 2.5. Sequence Analysis and Phylogenetic Tree Construction

Sanger sequencing was used for sequence validation, and the results were visualized with SnapGene software (version 6.0.2; Dotmatics, San Diego, CA, USA). The exon–intron structure of HSC70-3 was analyzed using the WormBase Exon–Intron Finder (http://www.wormweb.org/exonintron, accessed on 1 March 2025). The physico-chemical properties of HSC70-3 were predicted using ExPASy ProtParam (https://web.expasy.org/protparam/, accessed on 1 March 2025) based on the CDS. Amino acid composition, instability index, and aliphatic index were calculated to assess protein stability in vitro. The SMART tool (https://smart.embl.de/, accessed on 1 March 2025) identified conserved HSP70 domains. Phylogenetic analysis was performed using MEGA10 software with the maximum likelihood method and 1000 bootstrap replications to ensure statistical robustness. The constructed phylogenetic tree was visualized using the Chiplot online platform (https://www.chiplot.online/, accessed on 1 March 2025).

### 2.6. qRT-PCR

Primers specific to *hsc70-3* were designed based on its cDNA sequence using Primer Premier 6 software (PREMIER Biosoft, Palo Alto, CA, USA) and validated for specificity via BLAST analysis against the NCBI database (Table A1). The *P. xylostella rpl32* gene (Genome Database ID: *Px008022*) was used as the internal reference gene for normalization. Quantitative real-time PCR (qRT-PCR) was performed using the QuantStudio™ 6 Flex Real-Time PCR System (Applied Biosystems, Thermo Fisher Scientific, Waltham, MA, USA) in a 20 μL reaction mixture containing 10 μL GoTaq^®^ qPCR Master Mix (Promega Corporation, Madison, WI, USA), 0.2 μL of each primer (10 μM), 2 μL cDNA template, and 7.6 μL nuclease-free water. The thermal cycling conditions consisted of an initial denaturation at 95 °C for 3 min, followed by 40 cycles of 95 °C for 15 s, 60 °C for 15 s, and 72 °C for 30 s. Melt curve analysis (65–95 °C, with 0.5 °C increments every 5 s) was conducted to confirm amplification specificity. Each sample was analyzed in three biological replicates, with each biological replicate consisting of three technical replicates. The results of the technical replicates were averaged for each biological replicate, and these averaged values were used for statistical analysis. The relative expression levels of *hsc70-3* across developmental stages (E, L1, L2, L3, L4, P, and A) and tissues (FB, GCT, HD, MT, GUT, SK, RT) were quantified using the 2^−ΔΔCt^ method.

### 2.7. Protein Extraction and Western Blot

Total protein was extracted by homogenizing frozen samples on ice in 400 μL of radio immunoprecipitation assay (RIPA) lysis buffer (Beyotime Biotechnology, Shanghai, China) supplemented with a protease inhibitor (phenylmethylsulfonyl fluoride, PMSF; Solarbio, Beijing, China) and protease stabilizer (BestBio, Shanghai, China). The homogenate was gently vortexed and incubated at 4 °C for 30 min with intermittent shaking. The lysate was centrifuged at 14,000× *g* for 10 min at 4 °C to remove debris, and the supernatant was transferred into pre-chilled tubes. Protein concentration was determined using the Omni-Easy™ BCA Protein Quantification Kit (Epizyme Biomedical Technology Co., Ltd., Shanghai, China) following the manufacturer’s protocol. A 12.5% SDS-PAGE gel was prepared using the Omni-Easy™ One-Step PAGE Gel Preparation Kit (Epizyme Biomedical Technology Co., Ltd., Shanghai, China). Protein samples (20 μg per lane) were mixed with 5× loading buffer (Epizyme Biomedical Technology Co., Ltd., Shanghai, China), denatured at 100 °C for 10 min, briefly spun down, and then loaded onto the gel for electrophoresis. Electrophoresis was conducted at a constant voltage of 90 V for 20–30 min for protein stacking, followed by 130 V for approximately 80 min until complete separation. A pre-stained protein marker (Epizyme Biomedical Technology Co., Ltd., Shanghai, China) was included as a molecular weight reference. Proteins were transferred onto a polyvinylidene fluoride (PVDF) membrane (Millipore, Merck KGaA, Darmstadt, Germany) using a wet transfer system powered by the PowerPac™ Basic Power Supply (Bio-Rad, Bio-Rad Laboratories, Hercules, CA, USA) at 300 mA for 70 min at 4 °C. The membrane was blocked with TBST (Tris-buffered saline containing 0.5% Tween-20) supplemented with 3% skim milk at room temperature for 1 h and incubated overnight at 4 °C with the respective primary antibody (1:1000). Following three 10-min washes in TBST, the membrane was incubated with HRP-conjugated goat anti-rabbit IgG (1:5000, Affinity Biosciences, Changzhou, China) at room temperature for 2 h. Protein signals were detected using the Clarity™ Western ECL Substrate (Bio-Rad, Bio-Rad Laboratories, Hercules, CA, USA), and images were acquired using the Fusion FX5 imaging system (Vilber Lourmat, Collégien, France). Band intensities were quantified using ImageJ software (version 1.53a, National Institutes of Health, Bethesda, MD, USA), with relative protein levels normalized to β-actin expression.

Western blot (WB) analysis was conducted to assess the expression of HSC70-3 in various developmental stages (E, L1, L2, L3, L4, P, and A) and tissues (FB, GCT, HD, MT, GUT, SK, RT), as well as to investigate its secretion at feeding sites on radish leaves. Fourth-instar *P. xylostella* larvae reared on an artificial diet were starved for 4 h to standardize feeding behavior before being allowed to feed on radish leaves for 6 h. To restrict the feeding area, half of the leaf surface was covered with aluminum foil during feeding. Following feeding, leaves were excised, and non-fed areas were trimmed to match the fed regions, which served as the mechanical damage control (CK). The treated leaves were then directly pressed onto a PVDF membrane for protein transfer, and HSC70-3 detection was performed using standard WB procedures.

### 2.8. Immunofluorescent Localization of HSC70-3

To investigate the localization of HSC70-3 in feeding-related tissues, fourth-instar *P. xylostella* larvae reared on an artificial diet were dissected to collect the salivary glands, silk glands, Malpighian tubules, foregut, midgut, and hindgut. The dissected tissues were immediately fixed in 1 mL of 4% paraformaldehyde (Solarbio, Beijing, China) at 4 °C for 1 h. After fixation, tissues were washed three times with 0.01 M TBST (10 min each) and permeabilized in 0.2% Triton X-100 (Solarbio, Beijing, China) for 1 h at room temperature. Samples were then washed again with 0.01 M TBST (three times, 10 min each). Tissues were incubated with an anti-HSC70-3 polyclonal antibody (1:100, GenScript, Nanjing, China) at room temperature for 2 h or overnight at 4 °C with gentle shaking in a humidified chamber. After three washes with 0.01 M TBST (10 min each), Alexa Fluor 594-conjugated goat anti-rabbit IgG (1:500, Invitrogen, Thermo Fisher Scientific, Waltham, MA, USA) was applied and incubated for 2 h at room temperature in the dark. After three additional washes with TBST, the tissues were mounted on slides using DAPI Fluoromount-G (SouthernBiotech, Birmingham, AL, USA) and incubated in the dark for 10 min. Coverslips were placed over the samples and sealed with neutral balsam (Solarbio, Beijing, China). Fluorescence imaging was performed using a Leica TCS SP8 confocal laser scanning microscope (Leica Microsystems, Wetzlar, Germany). To further investigate HSC70-3 secretion at plant feeding sites, as described in Section 2.7, fourth-instar larvae were starved and then allowed to feed on radish leaves. The excised leaves were subsequently analyzed by immunofluorescence to detect HSC70-3 at the feeding sites.

### 2.9. Preparation of sgRNA and Embryo Microinjection

A 20 bp sgRNA targeting exon 1 of *hsc70-3* was designed using Benchling (https://www.benchling.com/, accessed on 1 March 2025). Potential off-target sites were predicted using Cas-OFFinder (http://www.rgenome.net/cas-offinder/, accessed on 1 March 2025) by screening the *P. xylostella* genome for mismatches. The sgRNA oligonucleotide was synthesized with a T7 promoter and an N_20_NGG sequence. To generate an in vitro transcription template, PCR amplification was performed using sgRNA-specific primers. Oligonucleotide sequences and primers for mutation detection were designed based on the genomic sequence (Table A1). The PCR product was purified by agarose gel electrophoresis, extracted using the Gel Extraction Kit (Omega Bio-Tek, Norcross, GA, USA), and used as a template for in vitro transcription. The sgRNA transcription reaction (10 µL) was assembled using the MEGAscript^®^ T7 High Yield Transcription Kit (Thermo Fisher Scientific, Ambion, Waltham, MA, USA) and contained 1 µL of 10× transcription buffer, 1 µL of enzyme mix, 1 µL ATP, 1 µL CTP, 1 µL UTP, 1 µL GTP, 2 µL purified PCR product, and 2 µL RNase-free ddH_2_O. The reaction mixture was incubated overnight at 37 °C. Following transcription, the reaction volume (10 µL) was diluted to 300 µL with nuclease-free water. An equal volume of phenol: chloroform: isoamyl alcohol (25:24:1) was added, gently mixed by inversion, and incubated at room temperature for 2 min. The mixture was then centrifuged at 14,000× *g* for 10 min at 4 °C, and the supernatant was transferred to a fresh tube. RNA was precipitated by adding 25 µL of 3 M sodium acetate and 750 µL of absolute ethanol, followed by incubation at −20 °C for 4 h. The RNA pellet was collected by centrifugation at 14,000× *g* for 15 min at 4 °C and washed twice with 1 mL of absolute ethanol. After air-drying, the pellet was resuspended in 20 µL of RNase-free water. RNA integrity was assessed via agarose gel electrophoresis, while concentration and purity were measured using a NanoDrop™ 2000 spectrophotometer (Thermo Scientific, Waltham, MA, USA). The RNA sample was subsequently stored at −80 °C for future use.

Following sgRNA and Cas9 protein preparation, the injection mixture was assembled by combining 150 ng/μL sgRNA and 300 ng/μL Cas9 in a nuclease-free PCR tube, followed by incubation at 37 °C for 30 min. A 3 μL aliquot was loaded into a sterile glass capillary needle (GD-1, Narishige, Tokyo, Japan) using a micropipette (Eppendorf Research^®^ plus, Eppendorf, Hamburg, Germany), and the needle tip was beveled under a stereomicroscope (SZX16, Olympus, Tokyo, Japan) to ensure precise penetration during microinjection. Freshly laid *P. xylostella* eggs (collected within 15 min of oviposition) were carefully transferred onto a clean microscope slide. Microinjections were performed using a stereomicroscope (SZX16, Olympus, Tokyo, Japan) in conjunction with a microinjection system (IM-300, Narishige, Tokyo, Japan). For each injection session, 300 embryos were injected. To minimize desiccation, each injection session was completed within 30 min per slide. Injected embryos were maintained at 25 °C for 24 h before being transferred to an artificial diet for further rearing. In cases where no mutations were detected after the first round of injections, a second round was performed, and the process was repeated until homozygous mutants were obtained.

### 2.10. Establishment of Mutant Strains

The *hsc70-3* mutant strain was established in our laboratory via CRISPR/Cas9 technology. Injected embryos developed into G_0_ larvae, which were reared to pupation and collected in sterile centrifuge tubes for subsequent analysis. Genomic DNA was extracted from G_0_ adults using the FastPure^®^ Cell/Tissue DNA Isolation Mini Kit (Vazyme Biotech, Nanjing, China), according to the manufacturer’s instructions. Genotyping was performed via PCR amplification with locus-specific primers, followed by Sanger sequencing at Sangon Biotech (Shanghai, China). Overlapping peaks in sequencing chromatograms indicated successful genome editing, and individuals carrying mutations were retained as the G_1_ generation. G_1_ individuals were crossed with wild-type adults to generate the G_2_ generation, and PCR-based genotyping was performed on G_2_ adults after mating to identify mutations. Self-crossing was carried out over multiple generations until homozygous mutants were established. Homozygous mutants were maintained for 4–5 generations, with mutation screening conducted in each generation to ensure genetic stability.

### 2.11. Insect Bioassays

To investigate the role of *hsc70-3* in the growth, development, and host adaptation of *P. xylostella*, newly hatched larvae (*n* = 120) from both the artificial-diet-reared wild-type (AD-WT) and *hsc70-3* mutant strains were collected and randomly assigned to six replicate groups per strain (20 larvae per group). Three groups per strain were reared on an artificial diet, while the remaining three groups were fed fresh radish seedlings. The artificial diet and radish seedlings were replenished every 48 h to ensure a continuous food supply. Mutant newly hatched larvae were transferred to the host plant for one generation (approximately two weeks), alongside the AD-WT, under identical rearing conditions. During the short-term host transfer experiment, biological parameters, including survival rate; larval weight on days 4, 5, and 6 after hatching; developmental duration from larva to pupa; pupal weight; and eclosion rate, were measured in parallel.

### 2.12. Statistical Analysis

All statistical analyses were performed using SPSS 21.0 (IBM Corp., Armonk, NY, USA), and data visualization was performed in GraphPad Prism 9.0 (GraphPad Software, San Diego, CA, USA). One-way ANOVA followed by Tukey’s post hoc test was used for multiple group comparisons, including qRT-PCR expression levels and protein expression levels across different developmental stages and tissues. Independent sample *t*-tests were applied to assess pairwise differences, such as larval weight, larval development duration, adult emergency rate, pupal weight, and larval survival rate between the wild-type and mutant strains. Statistical significance was set at *p* < 0.05. Results are presented as the mean ± standard error (SE).

## 3. Results

### 3.1. Structural and Physicochemical Characterization of HSC70-3

PCR amplification of the *hsc70-3* coding sequence (CDS) yielded a product of the expected size (1980 bp), confirming the successful cloning of the target gene. Exon–intron structure analysis showed that *hsc70-3* consists of two exons separated by one intron (Figure 1A). The sequence was validated via Sanger sequencing, and the results were visualized using SnapGene software (Figure 1B). The HSC70-3 consists of 659 amino acids, with a molecular weight of 73.1 kDa and a theoretical isoelectric point (pI) of 5.20. Sequence alignment revealed high similarity to HSP70 proteins in the UniProt database. Amino acid composition analysis showed that lysine (9.9%), aspartic acid (8.5%), and glutamic acid (8.3%) were the most abundant residues. The protein contains 111 negatively charged residues (aspartic acid and glutamic acid) and 92 positively charged residues (arginine and lysine). The instability index of HSC70-3 is 23.48 (below 40), while the aliphatic index is 84.1, indicating high stability in vitro. A highly conserved HSP70 domain was identified in HSC70-3, spanning nearly the entire sequence. A phylogenetic analysis based on HSC70 amino acid sequences from representative insects revealed that *P. xylostella* HSC70-3 clusters with Lepidopteran homologs, particularly *Spodoptera frugiperda* and *Manduca sexta*, indicating it is evolutionarily conserved within the order. Sequences from other insect orders formed distinct clades, while nematode and cladoceran sequences served as outgroups.

### 3.2. Developmental and Tissue-Specific Expression Patterns of hsc70-3

Specific primers were designed to amplify *hsc70-3* transcript for qRT-PCR analysis. qRT-PCR showed that *hsc70-3* expression was significantly upregulated in third-instar larvae (L3) compared to other developmental stages (*F*_(6,14)_ = 9.332, *p* < 0.001) (Figure 2A). Tissue-specific expression analysis revealed that *hsc70-3* was predominantly expressed in the fat body, followed by the gut contents and gut, while the lowest expression was in the silk gland (*F*_(5,12)_ = 248.095, *p* < 0.001) (Figure 2B). The results indicated that *hsc70-3* was preferentially expressed in the larval stage with a considerable amount in the gut.

The WB analysis was conducted to assess HSC70-3 protein expression across different developmental stages and tissues and the gut content where the regurgitant originates. Protein expression levels in eggs, fourth-instar larvae, pupae, and adults were significantly higher than those in third instar larvae, with no significant differences observed between these stages and second or first instar larvae (*F*_(6,14)_ = 4.007, *p* = 0.015) (Figure 3A). At the tissue level, HSC70-3 exhibited the highest protein abundance in the head and fat body, followed by the remaining tissues and silk gland (*F*_(6,14)_ = 7.039, *p* = 0.001) (Figure 3B). The results may indicate that as a molecular chaperone, the translated HSC70-3 protein was reallocated throughout the developmental stages and larval tissues with a small amount secreted into the extracellular gut content for regurgitation.

### 3.3. Tissue-Specific Localization of HSC70-3 in P. xylostella Larvae

Immunofluorescence analysis demonstrated the tissue-specific localization of HSC70-3 in *P. xylostella* larvae. DAPI (blue) stained nuclei, while red fluorescence signals indicated HSC70-3 localization. Strong fluorescence signals were detected in the silk gland and Malpighian tubule, where red fluorescence was evenly distributed across these tissues (Figure 4A). In the gut, HSC70-3 exhibited regional variation, with the strongest fluorescence signal detected in the foregut, whereas lower fluorescence intensity was observed in the midgut and hindgut (Figure 4B), conforming to the feature of an effector protein through the gut regurgitant.

### 3.4. Differential Expression Patterns of HSC70-3 in Larval Tissues Following a Short-Term Host Transfer

To further explore the role of HSC70-3 in *P. xylostella* adaptation to host plants, we examined its expression profile following a short-term host transfer from artificial diet to the radish seedlings in a laboratory strain of *P. xylostella*. The qRT-PCR was used to quantify *hsc70-3* expression, while the WB analysis was performed to assess HSC70-3 protein levels. qRT-PCR analysis revealed that *hsc70-3* was differentially expressed across various tissues of *P. xylostella*. The highest expression was detected in the silk gland, followed by the Malpighian tubule, while relatively low expression levels were observed in the fat body, head, gut, and remaining tissues (*F*_(5,12)_ = 11.527, *p* < 0.001) (Figure 5A). Notably, WB analysis revealed that HSC70-3 levels were elevated in the gut, silk gland, and the remaining tissues, whereas lower protein abundance was detected in the gut contents, Malpighian tubule, and head (*F*_(6,14)_ = 23.985, *p* < 0.001) (Figure 5B). The results revealed that HSC70-3 was likely secreted along with the gut regurgitant, except for the function as an intracellular molecular chaperone.

### 3.5. Release of HSC70-3 into Host Plant Tissues During P. xylostella Feeding

To verify the presence of HSC70-3 at insect feeding sites, in situ WB analysis was conducted. Strong HSC70-3 signals were detected at feeding sites (Attack, red arrow), while only minimal signals were present in control areas (CK, blue arrow) (Figure 6A). Immunofluorescence analysis further validated HSC70-3 localization at *P. xylostella* feeding sites. Strong red fluorescence signals were observed in damaged plant tissues (A1, A2, and A3), while control samples (CK1, CK2, and CK3) exhibited weak fluorescence (Figure 6B). The results confirmed that HSC70-3 was secreted into wounded plant tissue through larval gut regurgitant.

### 3.6. Functional Analysis of hsc70-3 in Larval Development and Host Adaptation in P. xylostella

To investigate the role of *hsc70-3* in larval development and host plant adaptation in *P. xylostella*, CRISPR/Cas9 technology was employed to generate the *hsc70-3* knockout mutant by targeting exon 2. Among the 600 injected eggs, 66 hatched (11%), of which 61 developed into adults (G_0_), and 9 exhibited gene editing (1.5%). After seven generations of inbreeding, a homozygous mutant line with a 12-bp deletion in exon 2 was established, leading to partial amino acid deletions. Bioassays on an artificial diet revealed no significant differences in larval performance between wild-type and ∆*hsc70-3* mutants (Figure 7A). However, when reared on radish seedlings, ∆*hsc70-3* mutants exhibited significantly lower body weights on day 4 (*t* = 3.502, *df* = 3.531, *p* = 0.030) and day 6 (*t* = 9.432, *df* = 3.799, *p* = 0.001), along with reduced pupal weights (*t* = 5.930, *df* = 2.352, *p* = 0.018) compared to AD-WT larvae (Figure 7B). Additionally, developmental duration was significantly prolonged in ∆*hsc70-3* mutants relative to AD-WT larvae (*t* = −4.950, *df* = 4, *p* = 0.008). By contrast, no significant differences were observed in adult emergence rates or larval survival between AD-WT and ∆*hsc70-3* mutant lines (Figure 7B). Collectively, these findings suggest that the compromised performance of ∆*hsc70-3* mutants is unlikely due to a direct impairment of intrinsic feeding capacity. Instead, the mutation appears to interfere with the ability of the insect to overcome host plant defenses, thereby reducing feeding efficiency and ultimately hindering growth and developmental processes on host plants.

## 4. Discussion

Over evolutionary timescales, herbivorous insects have evolved diverse feeding strategies and effector secretion systems to circumvent host plant defenses, thereby enhancing their adaptability [36,37,38]. The co-evolutionary “arms race” between plants and herbivores has driven the specialization of insect effectors, enabling them to suppress plant immune responses and enhance feeding success [3,37,39]. While insect saliva has been extensively studied in plant–insect interactions, particularly in piercing–sucking insects such as aphids and mirids, research on effectors present in the gut regurgitant of Lepidoptera remains poorly characterized [38,40,41]. This study demonstrates that HSC70-3, traditionally recognized as a molecular chaperone involved in cellular stress responses, may function as an effector in *P. xylostella*, revealing a novel mechanism of host adaptation.

Short-term host transfer of *P. xylostella* from an artificial diet to radish seedlings resulted in significant changes in HSC70-3 expression across various larval tissues. Despite the relatively low mRNA levels in gut tissues, the corresponding protein abundance remained high, suggesting potential posttranslational regulation or tissue-specific variations in protein stability. The observed discrepancy between mRNA and protein levels may result from multiple regulatory mechanisms, as transcript levels alone often fail to precisely reflect protein abundance [42,43,44]. Interestingly, although HSC70-3 protein levels were relatively high in gut tissues, its abundance in the gut contents was markedly reduced. This spatial discrepancy indicates that HSC70-3 is likely retained within gut epithelial cells rather than being passively released into the gut lumen. This retention is likely linked to its role as a molecular chaperone, where it facilitates protein folding, stabilization, and intracellular transport [24]. Consequently, only a small proportion of HSC70-3 is secreted into plant wounds through gut regurgitant during feeding. Therefore, the secretion of HSC70-3 into plant tissues is likely a regulated process, consistent with its well-known functions in protein stabilization and stress responses [24]. Gut regurgitant, as a critical interface between insect and host plant, could serve as a potential vehicle for the delivery of HSC70-3, which supports the notion that HSC70-3 may contribute to insect–plant interactions. Furthermore, the significantly higher HSC70-3 levels in FB tissues relative to its mRNA levels suggest that the fat body may function as a reservoir or transport hub for HSC70-3. As a key tissue involved in protein metabolism, the fat body plays a pivotal role in insect environmental adaptation [45,46]. The elevated expression of HSC70-3 in this tissue may contribute to an enhanced immune capacity, facilitating adaptation to physiological stress imposed by the host plant. In conclusion, following short-term host plant transfer, tissue-specific expression patterns of HSC70-3 exhibited noticeable changes, albeit modest in magnitude. This may be attributed to the focus on short-term responses, where changes may not have fully manifested in the early stages following transfer. Alternatively, the host plant transfer may have induced an initial stress response that was not fully captured in the present study. Future studies should explore the long-term effects of host plant feeding on HSC70-3 expression, especially by incorporating a control group continuously feeding on host plants. This would enable a more comprehensive comparison of HSC70-3 expression between short-term host plant transfer and continuous feeding on host plants. Moreover, the observed changes in HSC70-3 expression could result from stress induced by host plant transfer or plant-derived challenges, potentially contributing to the increase in HSC70-3 levels. Further investigations are needed to elucidate the mechanisms underlying these stress responses and their impact on HSC70-3 expression.

Immunofluorescence and in situ WB analyses confirmed that HSC70-3 is secreted and accumulates at plant wound sites during *P. xylostella* larval feeding. This finding suggests that HSC70-3 may play a dual role, functioning not only in insect stress responses but also in the modulation of plant defense mechanisms. As a multifunctional molecular chaperone, HSC70-3 facilitates protein folding and stress regulation within insect cells and may interact with plant membrane receptors or signaling molecules, thereby influencing host immune responses [47,48]. Previous studies have presented indirect evidence supporting the extracellular function of HSC70-3 in plant–insect interactions [32]. In *N. lugens*, the HSP70 family member NlHSC70-3 significantly suppresses Flg22-induced ROS accumulation and downregulates the expression of key pathogenesis-related genes (*NbPR1*, *NbPR2*, *NbPR3*, and *NbPR4*) in *Nicotiana benthamiana*, thereby interfering with rice immune responses and promoting insect feeding [32]. The identification of HSC70-3 as a potential effector in *P. xylostella* provides compelling evidence that gut regurgitant plays a crucial role in insect adaptation to host plants. This finding broadens the current understanding of effector-mediated interactions in Lepidoptera and highlights a novel avenue for investigating the interplay between gut-derived secretions and plant signaling pathways.

CRISPR/Cas9-mediated knockout of *hsc70-3* provided functional insights into the essential role of HSC70-3 in *P. xylostella* larval adaptation. Mutant larvae exhibited significantly reduced body weight, delayed development, and decreased pupal weight when reared on radish seedlings, suggesting that *hsc70-3* facilitates host plant adaptation by modulating plant defense responses and optimizing nutrient assimilation. However, it is important to note that plant defense was not directly measured in this study. These findings are based on the observed developmental impairments in mutant larvae, which may be linked to an impaired ability to modulate plant defense. The previous studies on *H. zea*, in which the effector-mediated suppression of plant immunity promotes larval growth and survival, support this potential role of HSC70-3 in overcoming plant resistance mechanisms [49]. Notably, *hsc70-3* knockout larvae exhibited no significant developmental defects when reared on artificial diets, supporting the notion that *hsc70-3* primarily modulates plant-induced defense responses rather than directly affecting insect metabolism or development. This finding highlights the critical role of effectors in overcoming plant resistance mechanisms, although a direct measurement of plant defense was not conducted in this study. Previous studies have shown that HSC70 family proteins are involved in nutrient metabolism and stress responses in various insect species [48,50]. Accordingly, we do not exclude the possibility that HSC70-3 may influence larval development by modulating the absorption or utilization of plant-derived nutrients. Further investigation is warranted to determine whether HSC70-3 contributes to nutrient acquisition in addition to its role in suppressing plant defense responses. Although only three biological replicates were used for survival and adult emergence rate assays, this limitation was due to the restricted number of eggs produced by CRISPR-edited lines and the need to maintain their genetic stability across generations. To ensure reliable interpretation, we additionally measured larval weight, developmental duration, and pupal weight, which are widely recognized as robust indicators of insect fitness. These complementary data consistently supported the biological role of *hsc70-3* in host adaptation.

Although this study presents strong evidence supporting the role of HSC70-3 as an effector in *P. xylostella*, the precise molecular mechanisms underlying its role remain to be fully elucidated. Growing evidence indicates that herbivorous insects utilize effectors to interfere with plant defense pathways via multiple mechanisms [1,51,52]. While some insect effectors have been shown to interact with specific plant receptors or signaling molecules to suppress plant defense pathways, whether HSC70-3 adopts a similar strategy in *P. xylostella* remains unknown. Future research should investigate the specific interactions between HSC70-3 and plant immune receptors or signaling pathways to elucidate its role in host adaptation. The identification of HSC70-3 as an immune-suppressing effector in *P. xylostella* suggests novel strategies for pest control. Specifically, plant defense responses could be measured using qRT-PCR to analyze the expression of defense-related genes or by assessing ROS accumulation and pathogenesis-related proteins to evaluate the impact of HSC70-3 on plant immunity. If HSC70-3 significantly attenuates plant defenses, its targeted inhibition could enhance crop resistance towards herbivorous insects. Furthermore, identifying the receptors or downstream targets of HSC70-3 in plants could facilitate the development of genetically modified pest-resistant crops, thereby improving plant resilience against *P. xylostella*.

## 5. Conclusions

In this study, we identified and characterized HSC70-3 in the gut regurgitant of *P. xylostella* and demonstrated that it possesses a signal peptide and is secreted into plant wound sites during larval feeding. Immunofluorescence and in situ WB further confirmed its accumulation in damaged plant tissues. Short-term host transfer experiments revealed distinct tissue-specific expression dynamics of *hsc70-3*, indicating its differential regulation at both transcriptional and post-translational levels in response to plant-derived stressors. CRISPR/Cas9-mediated knockout of *hsc70-3* significantly impaired larval growth, prolonged development, and reduced pupal weight on host plants, further highlighting its functional role in host adaptation.

These findings provide direct evidence that HSC70-3 functions as an effector in the interaction between *P. xylostella* and host plants. Further research is needed to identify its molecular targets in host plants, which could enhance our understanding of insect-mediated modulation of plant immunity and inform pest management strategies.

## Figures and Tables

**Figure 1 insects-16-00489-f001:**
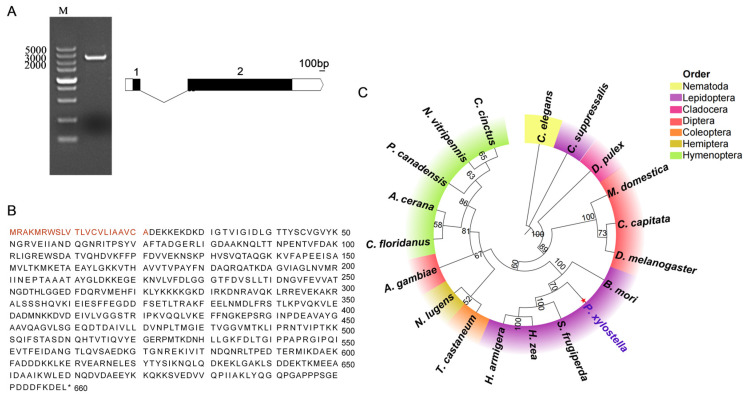
Molecular characterization and phylogeny of HSC70-3. (**A**) PCR amplification and exon–intron structure of *hsc70-3* Lane M: DNA marker with fragment sizes indicated on the left (bp). The schematic representation on the right illustrates the exon–intron structure of *hsc70-3*, where black boxes denote coding exons, white boxes indicate untranslated regions (UTRs), and the curved line represents the intron. The scale bar corresponds to 100 bp. (**B**) Amino acid sequence of HSC70-3, with the signal peptide highlighted in orange. (**C**) Phylogenetic tree of HSC70 family proteins from *P. xylostella* and representative species across Nematoda, Lepidoptera, Cladocera, Diptera, Coleoptera, Hemiptera, and Hymenoptera. The tree was generated in MEGA X (version 10.2.6; Mega Limited, Auckland, New Zealand) using the maximum likelihood method with 1000 bootstrap replicates. Taxonomic groups are shown in different colors, and the *P. xylostella* HSC70-3 sequence is marked in purple. Protein sequences were obtained from the NCBI database. The sequence details are as follows: *Caenorhabditis elegans* (NP_001370435.1), *Chilo suppressalis* (BAE44308.1), *Bombyx mori* (AEI58998.1), *Spodoptera frugiperda* (QOW03265.1), *Helicoverpa zea* (XP_047039469.1), *Helicoverpa armigera* (AEB26351.1), *Daphnia pulex* (XP_046438519.1), *Musca domestica* (XP_005184291.1), *Ceratitis capitata* (XP_004535076.1), *Drosophila melanogaster* (NP_001285139.1), *Anopheles gambiae* (XP_061503529.1), *Tribolium castaneum* (XP_008200986.2), *Nilaparvata lugens* (AQP31362.1), *Camponotus floridanus* (XP_025269686.1), *Apis cerana* (XP_061928184.1), *Polistes canadensis* (XP_014603421.1), *Nasonia vitripennis* (XP_001604460.2), and *Cephus cinctus* (XP_015596735.1).

**Figure 2 insects-16-00489-f002:**
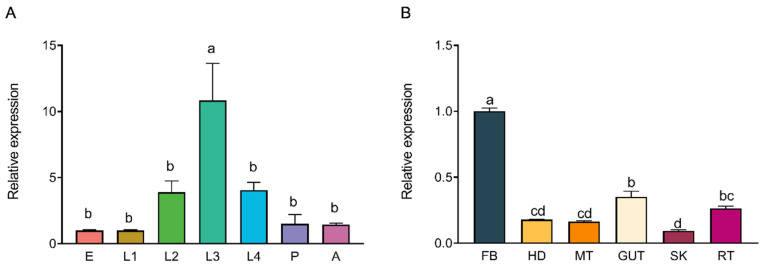
Developmental and tissue-specific expression patterns of *hsc70-3*. (**A**) qRT-PCR analysis of *hsc70-3* expression across different developmental stages of *P. xylostella*. E: egg; L1: first instar; L2: second instar; L3: third instar; L4: fourth instar; P: pupa; A: adult. (**B**) qRT-PCR analysis of *hsc70-3* expression across different tissues of *P. xylostella* from fourth-instar larvae. FB: fat body; HD: head; MT: Malpighian tubule; GUT: gut; SK: silk gland; RT: remaining tissues. Statistical analysis was conducted using one-way ANOVA followed by Tukey’s post hoc test (*p* < 0.05). Different letters indicate statistically significant differences among tissues. Data are presented as mean ± standard error (SE) from three biological replicates.

**Figure 3 insects-16-00489-f003:**
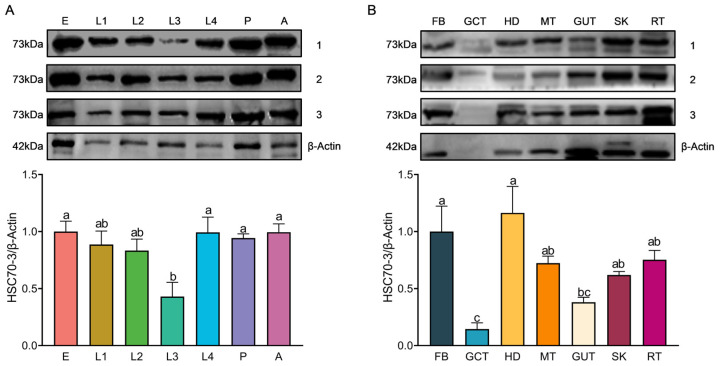
Developmental and tissue-specific expression patterns of HSC70-3. (**A**) WB analysis of HSC70-3 protein expression across different developmental stages of *P. xylostella*. E: egg; L1: first instar; L2: second instar; L3: third instar; L4: fourth instar; P: pupa; A: adult. (**B**) WB analysis of HSC70-3 protein expression in different tissues of *P. xylostella*. FB: fat body; GCT: gut contents; HD: head; MT: Malpighian tubule; GUT: gut; SK: silk gland; RT: remaining tissues. Statistical analysis was conducted using one-way ANOVA followed by Tukey’s post hoc test (*p* < 0.05). Different letters indicate statistically significant differences among tissues. Data are presented as mean ± standard error (SE) from three biological replicates.

**Figure 4 insects-16-00489-f004:**
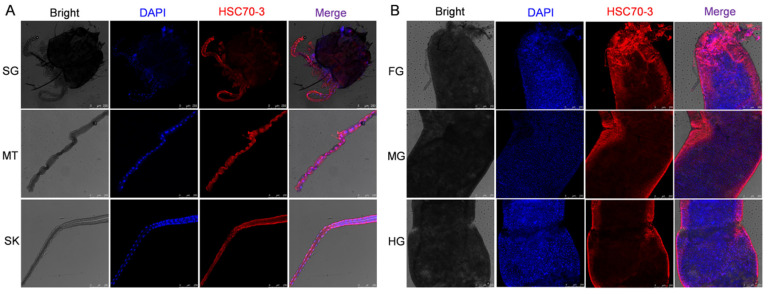
Immunofluorescence localization of HSC70-3 in various larval tissues. (**A**) Immunofluorescence analysis of HSC70-3 localization in the salivary gland (SG), Malpighian tubule (MT), and silk gland (SK). (**B**) Immunofluorescence analysis of HSC70-3 localization in the foregut (FG), midgut (MG), and hindgut (HG). Nuclei were stained with DAPI (blue), and HSC70-3 signals were detected in red. Scale bar = 250 µm.

**Figure 5 insects-16-00489-f005:**
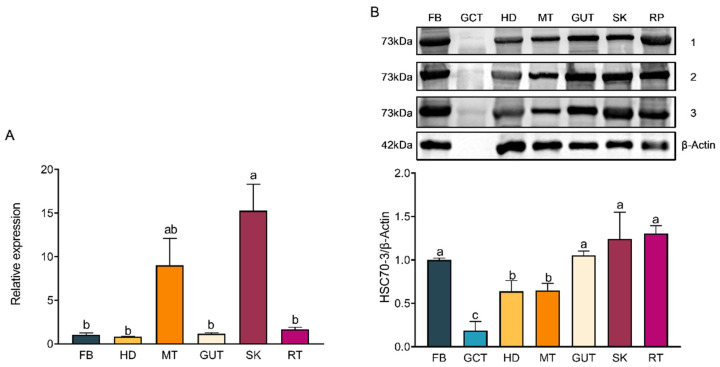
*hsc70-3* and protein expression patterns in larval tissues following short-term host transfer (**A**) WB analysis of HSC70-3 protein levels across different tissues. β-Actin served as a loading control. FB: fat body; HD: head; MT: Malpighian tubule; GUT: gut; SK: silk gland; RT: remaining tissues. All tissues were collected from a laboratory strain of *P. xylostella* following a short-term host transfer from artificial diet to radish seedlings. Band intensities were quantified using ImageJ and normalized to β-Actin. (**B**) qRT-PCR analysis of *hsc70-3* mRNA expression across different tissues. Statistical analysis was conducted using one-way ANOVA followed by Tukey’s post hoc test (*p* < 0.05). Different letters indicate statistically significant differences among tissues. Data are presented as mean ± standard error (SE) from three biological replicates.

**Figure 6 insects-16-00489-f006:**
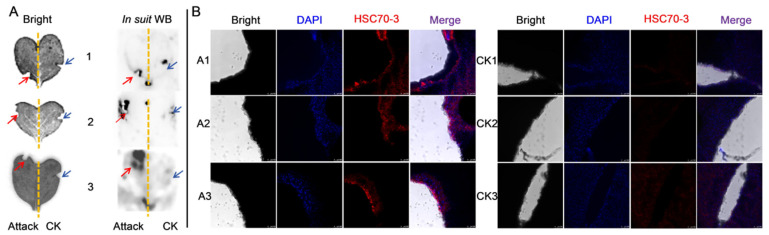
HSC70-3 localization in host plant tissues following larval feeding. (**A**) In situ WB analysis of HSC70-3 expression. Attack: *P. xylostella* feeding treatment, indicated by red arrows; CK: Mechanical damage treatment, indicated by blue arrows. 1–3: Three biological replicates. (**B**) Immunofluorescence analysis of HSC70-3 localization at feeding sites. A1–A3: *P. xylostella* feeding treatment (three biological replicates); CK1–CK3: Mechanical damage treatment (three biological replicates). DAPI (blue) stains nuclei, and HSC70-3 is shown in red. Merged images illustrate HSC70-3 localization within plant tissues. Scale bar = 100 µm.

**Figure 7 insects-16-00489-f007:**
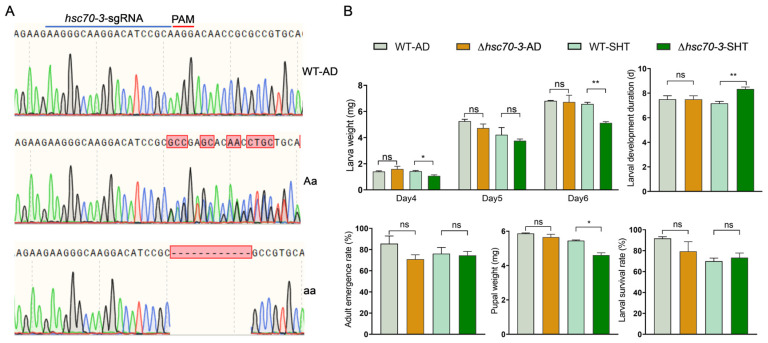
Effects of *hsc70-3* knockout on larval fitness to host plant. (**A**) Sanger sequencing validation of *hsc70-3* knockout mutations. WT-AD: wild-type *P. xylostella* reared on an artificial diet; Aa: heterozygous; aa: homozygous. The red box highlights the deleted 12 bp region in heterozygous mutants, while the dashed box indicates the corresponding deletion in homozygous mutants. (**B**) Effects of *hsc70-3* knockout on larval growth and development. Larval weight was measured on days 4, 5, and 6 (top left). Larval development duration (top right), adult emergence rate (bottom left), pupal weight (bottom middle), and larval survival rate (bottom right) were also recorded. Δ*hsc70-3*-AD: *hsc70-3* knockout line reared on an artificial diet; WT-SHT: Artificial diet strain with short-term host transfer; Δ*hsc70-3*-SHT: *hsc70-3* knockout line reared under short-term host transfer conditions. Data are presented as mean ± standard error (SE). Statistical differences between groups were analyzed using an independent-sample *t*-test (*p* < 0.05: *, *p* < 0.01: **, ns: not significant).

## Data Availability

The data presented in this study are available upon reasonable request from the corresponding author.

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
