# Peer review of "HSC70-3 in the Gut Regurgitant of Diamondback Moth, *Plutella xylostella*: A Candidate Effector for Host Plant Adaptation"

_insects, 2025, doi:10.3390/insects16050489_

Round 1

Reviewer 1 Report

Comments and Suggestions for Authors

How herbivorous insects overcome the defensive responses of hosts is a hot research topic internationally, and researches in this field help to deepen the understandings of insect-plant coevolution. Here, the authors focused on the gut regurgitant, which is secreted into plants during feeding of Plutella xylostella (a serious crop pest) and potentially contains certain effectors related to host adaptation. I read throughout the text, and in my sense the experiments were very well designed. A potential effector, HSC70-3, was identified, which was showed to play a crucial role in the insect's adaptation to plant. Functions of HSPs in this field were little learned before, and the findings of this effector are of great values.

I have only a few comments on this paper. In terms of research methodology, the number of replicates used for calculating adult emergence rates and larval survival rates seems somewhat insufficient (Fig. 7B): each treatment only has three replicates, with only 20 insects in per replicate. To my opinion, for groups having 20–30 insects, at least five replicates (groups) would be necessary to yield reliable results. On the other hand, as larval weight, developmental duration, and pupal weight can effectively reflect the fitness of insects (Fig. 7B), the limitations in the two percentage-based metrics (emergence and larval survival rates) would not undermine the conclusions.

I do not learn much about the stories happening at interface of herbivorous insects and plants. Yet as I think about them in the context of nutritional physiology, I imagine whether the HSC70-3 is possibly related with utilization of certain nutrients. In other words, reductions in larval and pupal weight in Δhsc70-3-SHT insects (Fig. 7B) might be due to the reduced absorption/utilization of certain nutrients, but not to the plants’ defense responses. Could HSPs contribution to the utilization of nutrients from plants?

L419-421: “… were significantly higher in egg, fourth instar larvae, pupae, and adults”: According to Fig. 3A, though the protein abundances in L2 and L3 are lower, they do not differ significantly with those in egg, fourth instar larvae, pupae, and adults. I suggest rewriting this sentence.

L467: β-Actin served as…

L347: “larval weight on days 4 and 5”: it should be done on “days 4, 5 and 6”?

L503: larvae (Figure 8A) => larvae (Figure 7B).

Reviewer 2 Report

Comments and Suggestions for Authors

Dear editor and authors,

In this manuscript, Qiao et al. examined HSC70-3 in diamond moth gut could be acted as a candidate effector for host plant adaptation. The authors used qPCR and western blot to show the gene expression at different development stage and tissues. With immunofluorescent localization, they also found the protein accumulates at the feeding sites on host plants. Expression changed after short-term transfer from artificial diet to host plants. Knockout of hsc70-3 significantly impacted larval fitness on host plants but not on artificial diet, indicating its potential involvement in host adaptation. The manuscript is well written, and the data presentation is clear. However, several aspects require clarification, and I have provided comments and suggestions to further improve the quality of the manuscript.

Introduction

It is well written, I only have one question here about the order, based on my understand, HSP or HSC could be acted as effectors in enhancing the plant defence, why the authors made the paragraph 2 and 3 as separate sections. Is it due to other important function or remain largely unexplored for HSP or HSC compared to other effectors mentioned in Paragraph 2? I think the logic here should be a bit clear. Apart from that, I only have minor revision for this part.

Materials and Methods

The M&M is also very clear. The authors have worked very hard on this point. However, I would have appreciated some information or section added and moved one section backwards. Additionally, the number of experimental replicates should be specified. A separate section of ‘a short-term host transfer’ should be clear marked in the methods without combing with other section.

Line 125: for how many generations or how long time?

Line 144: what is PBS? Is it Phosphate-buffered saline?

Line 138-156: this section should be moved backwards and combined with section 2.8.

Line 218: how did the authors analyse the data using three technique controls? Average them as one biological control?

Line 221: if the authors moved line 138-156, then abbreviation for tissues should be explained here.

Section 2.5 It would be clear to add where the western has been used in the experiment which I assumed apart from feeding sites, protein and western has also been used to detect HSC 70-3 in different development stages and tissues.

Line 281-285: As this has been mentioned before, it might be good to refer to previous section rather than giving the same description again or the authors could add ‘an artificial diet were used for in situ protein detection and immunofluorescent localization’ in line 255.

Line 324: how many embryos has been used for microinjection?

Section 2.10: be specific about what data was used for one-way ANOVA or t-test.

Results:

The results are clearly presented. I really like the authors used qPCR and Western together to show the expression in different development stage and tissues. As the information in method of short-term host plant transfer was not clearly marked, I am not sure how the authors did the experiment and how many generations or time meant short-term here as well as the development stage for transfer. My main suggestion was section 3.4 and 3.6. In the future study, the authors could consider adding one control as diamondback moth feeding consistently on the plant hosts.  I could see the authors in Fig 2 showed that the diamondback moth fed on the artificial diet as a control and the changes seemed not massive in protein level after transfer. And I would suggest the authors could consider including the treatment with consistently feeding on the host plants in their future study or clarify why they have not included it. As the insects were maintained on AD, the high expression might be caused by the host switch (could be due to plant-derived challenges but it also could be treated as stress which might cause hsp increase) or the plant itself. So did section 3.6.

Line 403: The authors should check the data again to make sure they use the same format such as whether to use Capital P or lowercase p.

Section 3.1: All these tools used to analyse the sequence and the plot phylogenetic tree should be added to the method section into 2.10.

Discussion

In my opinion, the authors rushed the discussion a bit. The discussion lacks mechanistic explanations with previous studies.

Line 544-546: I don’t understand this idea. What did the authors mean about the limited content of HSC70-3? And what was the relationship with a molecular chaperone here?

Line 558-562: add reference to these two sentences.

Line 574-575 and line 580-581: I am unsure about this as the plant defense has not been measured in this study.

Minor revision

Line 19: add ‘diamondback moth’ before feeding sites

Line 20: delete ‘function to’

Line 22: on the same host plants

Line 23: add main nutrients for an artificial diet such as ‘an artificial diet containing rich protein and carbohydrate’

Line 27: add content after ‘crop protection by…’

Line 34: add the host plant academic name Nanpan Prefecture

Line 67: I feel the authors need a sentence to point out the main content for this paragraph which could be moved the sentence in line 78-79 to front as ‘Previous findings highlight the essential role of effectors in overcoming plant defenses and facilitating insect adaptation. For example, Glucose oxidase (GOX)….’

Line 92: As the authors did not mention the study in detail with reference 23. It might be clear to mention briefly this study is done in other species such as ‘also shown in other insect species [23].’

 Line 109: it is too vague here to show the importance about this study. Add more information here to show why it is important to study the function in Plutella xylostella such as due to its important economic significant or recent quick spread in XX?

Line 120: the last sentence might be more suitable for discussion.

Line 152: twenty not ‘20’

Line 358-361: Delete this sentence as this information has been mentioned in method.

Line 412: add four-instar larvae in figure legend here to show the tissues were taken from larvae

Line 467: delete ‘。’

 Line 592: add what kind of resistance as ‘its targeted inhibition could enhance crop resistance towards …’

Line 601: hsc70-3 in italic

Comments on the Quality of English Language

Can be improved by minor modifications.

Round 2

Reviewer 2 Report

Comments and Suggestions for Authors

Dear authors and editor,

The authors have thoroughly responded to my concerns brought up in review. All prior comments and suggested edits have been addressed appropriately.